# Cellular and Molecular Biology of Mitochondria in Chronic Obstructive Pulmonary Disease

**DOI:** 10.3390/ijms25147780

**Published:** 2024-07-16

**Authors:** Chin-Ling Li, Shih-Feng Liu

**Affiliations:** 1Department of Respiratory Therapy, Kaohsiung Chang Gung Memorial Hospital, Kaohsiung 833, Taiwan; musquito16@cgmh.org.tw; 2Division of Pulmonary and Critical Care Medicine, Department of Internal Medicine, Kaohsiung Chang Gung Memorial Hospital, #123, Ta-Pei Road, Niaosong District, Kaohsiung 833, Taiwan; 3College of Medicine, Chang Gung University, Taoyuan 333, Taiwan

**Keywords:** chronic obstructive pulmonary disease, cellular and molecular biology of mitochondria, antioxidants and mitochondrial biogenesis, mitochondrial biomarkers

## Abstract

Chronic obstructive pulmonary disease (COPD) is a progressive respiratory disorder characterized by enduring airflow limitation and chronic inflammation. Growing evidence highlights mitochondrial dysfunction as a critical factor in COPD development and progression. This review explores the cellular and molecular biology of mitochondria in COPD, focusing on structural and functional changes, including alterations in mitochondrial shape, behavior, and respiratory chain complexes. We discuss the impact on cellular signaling pathways, apoptosis, and cellular aging. Therapeutic strategies targeting mitochondrial dysfunction, such as antioxidants and mitochondrial biogenesis inducers, are examined for their potential to manage COPD. Additionally, we consider the role of mitochondrial biomarkers in diagnosis, evaluating disease progression, and monitoring treatment efficacy. Understanding the interplay between mitochondrial biology and COPD is crucial for developing targeted therapies to slow disease progression and improve patient outcomes. Despite advances, further research is needed to fully elucidate mitochondrial dysfunction mechanisms, discover new biomarkers, and develop targeted therapies, aiming for comprehensive disease management that preserves lung function and enhances the quality of life for COPD patients.

## 1. Introduction

Chronic obstructive pulmonary disease (COPD) is a relentlessly progressing global health issue, marked by persistent airflow obstruction and long-lasting respiratory symptoms [1,2]. Observations show that the prevalence of COPD is escalating worldwide, posing a substantial load on healthcare infrastructures and significantly affecting patients’ quality of life [3,4]. Though COPD has conventional links to cigarette smoking, it also intertwines a complex mix of genetic susceptibilities, environmental factors, and intricate molecular mechanisms that contribute to its pathogenesis [5,6]. Recent findings spotlight the relevance of mitochondrial dysfunction in perpetuating COPD development and propelling disease evolution [7,8].

Mitochondria, aptly called the cell’s powerhouses, are integral to several cellular processes, including metabolism, energy synthesis, and signal transduction [9,10]. In addition to these core functions, current research emphasizes the fundamental role mitochondria play in maintaining cellular equilibrium and managing stress responses [11,12]. Emerging research pinpoints mitochondrial dysfunction as a characteristic attribute in COPD pathology, implicated in diverse cellular operations such as inflammation, oxidative stress-induced damage, and apoptosis—programmed cell death [13,14].

Recognizing the nuances of cell biology, specifically the function and mechanisms of mitochondria in COPD, is crucial to deciphering the disease’s convoluted progression and identifying plausible therapeutic strategies and targets [15,16]. This review strives to provide a holistic depiction of the current understanding of mitochondrial dysfunction’s role in COPD, with a focus on structural alterations in mitochondria, the associated cellular repercussions, and potential therapeutic strategies aimed at promoting mitochondrial health restoration [17,18].

The aim is to untangle the complex relationship between mitochondrial biology and COPD, fostering a comprehensive understanding from which innovative therapeutic interventions and improved COPD management strategies can be developed [19,20]. Emphasizing the role of mitochondria in COPD underscores the potential of exploring mitochondria-targeted therapeutics and redox-based interventions as paths for enhancing patient outcomes and halting disease progression.

Lastly, future investigative efforts should concentrate on the prominent interactions between inflammation, oxidative stress, genetic predispositions, and environmental influences, which could instigate revolutionary transformations in COPD management and therapeutic approaches. A deeper understanding of mitochondrial dysfunction’s role in COPD could fuel the advancement of personalized medicine strategies aimed at tackling the disruption of cellular homeostasis. This could move the focus away from merely symptomatic treatment and towards specific disease modification, thereby offering hope for efficacious management of this chronic and debilitating condition.

## 2. Mitochondrial Dysfunction in COPD (Table 1)

Long anchored in the realm of pulmonary ailments, COPD—marked by chronic inflammation, excessive oxidative stress, and compromised lung function—is increasingly being recognized as a systemic disease with implications profound enough to affect the body beyond the confines of the respiratory system [21,22]. Integral to this narrative are the mitochondria, the dynamic entities within cells that orchestrate energy production, thereby ensuring cellular homeostasis. Their role in the pathogenesis of COPD is becoming increasingly significant.ijms-25-07780-t001_Table 1Table 1Therapeutic Targets for Mitochondrial Dysfunction in COPD.Therapeutic ApproachMechanism of ActionAntioxidant TherapyReduction in oxidative stress through neutralizing ROSMitochondrial Biogenesis InducersActivation of PGC-1α to promote mitochondrial functionMitochondrial Quality Control EnhancersEnhancement of mitophagy and activation of UPRmtModulators of Mitochondrial DynamicsRegulation of mitochondrial fusion and fission processesRole of Peptides of Mitochondrial OriginHumanin has demonstrated protective effects against oxidative stress-induced cell damage


### 2.1. Oxidative Stress, Inflammation, and Mitochondrial Dysfunction

The association between COPD, oxidative stress, and chronic inflammation is well-established. The excessive production of reactive oxygen species (ROS) within the body leads to oxidative damage and impaired mitochondrial function [23]. This damage exacerbates mitochondrial disruption, specifically targeting the electron transport chain (ETC) vital for energy production. Additionally, chronic inflammation in COPD, fueled by cellular immune responses and pro-inflammatory mediators, further impairs mitochondrial function, creating a vicious cycle of damage and disease progression [18,24,25].

### 2.2. Altered Mitochondrial Morphology and Dynamics

Nuanced findings from ultrastructural examinations reveal profound alterations in the morphology and dynamism of mitochondria within individuals afflicted with COPD [16]. Manifestations range from mitochondrial fragmentation and elongation to disruptions in the critical process of mitochondrial biogenesis, implying a disrupted equilibrium of mitochondrial fission and fusion. These disturbances, no less than a pervasive disorder of mitochondrial dynamics, lead to dyshomeostasis of cellular function, impacting ATP production, calcium balance, and mitochondria turnover—all of which are integral to maintaining cell health [26,27].

### 2.3. Impaired Mitochondrial Respiratory Function

The deleterious effect of COPD extends to compromising mitochondrial respiratory function, leading to an inefficiency in energy production [28,29]. Evidence supports a reduction in the activity of respiratory chain complexes, particularly complex I and IV, within COPD patients. This compromised mitochondrial respiration has significant repercussions; notably, it is associated with muscle weakness, exercise intolerance, and an array of systemic manifestations that are characteristic of COPD [30,31].

### 2.4. Mitochondrial DNA Damage and Mutations

Given its proximity to ROS production epicenters within the mitochondria, mtDNA is particularly susceptible to oxidative damage [32]. Studies reveal an alarming increase in damage and mutations to mtDNA within COPD patients. These modifications have a fallout that jeopardizes mitochondrial protein synthesis and respiratory functionality. Furthermore, such mtDNA-related aberrations find relations with systemic COPD manifestations like skeletal muscle dysfunction and cardiovascular comorbidities, expanding the impact of the disease [33,34,35].

Gaining a comprehensive understanding of the complex interplays between mitochondrial dysfunction and COPD is pivotal for the progression of targeted therapeutic strategies aiming to restore mitochondrial health and limit disease progression [36,37]. This potential shift from symptomatic management to cellular normalization could broaden the spectrum of viable coping strategies for COPD like never before. Tackling oxidative stress, limiting inflammation, and rejuvenating the functionality of mitochondria collectively offers a pathway to transform and redefine the COPD treatment narrative. This potential leap forward holds promise for significantly improving patient outcomes while finding ways to halt disease progression.

## 3. Structural and Functional Changes in Mitochondria (Table 1)

Mitochondria, highly dynamic organelles responsible for cellular energy production, undergo significant structural and functional alterations in the milieu of COPD [1,38]. These perturbations have far-reaching implications on cellular function, positioning them at the heart of COPD pathogenesis.

### 3.1. Altered Mitochondrial Morphology

In-depth ultrastructural analyses have revealed dramatic changes in mitochondrial shapes and sizes in patients afflicted with COPD [28]. This includes mitochondrial fragmentation, swelling, and compromise in the integrity of cristae—internal compartments vital to mitochondrial function. This disruption in mitochondrial morphology is closely associated with compromised mitochondrial function and a disturbance in cellular equilibrium. While the exact mechanisms fueling these morphological alterations in mitochondria remain a subject of ongoing research, probable culprits include oxidative stress, inflammation, and irregularities in mitochondrial dynamics [39].

### 3.2. Impaired Mitochondrial Dynamics

Mitochondrial dynamics encompass the fundamental processes of fission and fusion, which are crucial for maintaining mitochondrial homeostasis. Within the sphere of COPD, irregularities in these dynamics fuel mitochondrial fragmentation and impair quality-control mechanisms, leading to a decline in ATP production, disturbed calcium balance, and increased susceptibility to apoptosis—programmed cell death [40]. This points towards the possibility of manipulating mitochondrial dynamics as a new avenue for therapeutic intervention.

### 3.3. Mitochondrial Respiratory Chain Dysfunction

The mitochondrial respiratory chain, composed of five enzymatic complexes, underpins oxidative phosphorylation and ATP synthesis—the cell’s primary energy production machinery. Disturbances in the function of this respiratory chain have been reported in COPD, characterized by reduced activity levels of complex I and IV [6,41]. This dysfunction leads to decreased ATP production and the increased reactivity of oxygen species (ROS), contributing to cellular inflammation and damage that are defining features of COPD.

### 3.4. Disrupted Mitochondrial Biogenesis

Mitochondrial biogenesis—the process responsible for the creation of new mitochondria—is governed by an interconnected network of nuclear and mitochondrial factors [42]. In the narrative of COPD, this biogenesis process is disrupted, negatively impacting mitochondrial turnover and function. This disruption is further amplified by a reduced expression of regulators like the peroxisome proliferator-activated receptor gamma coactivator 1-alpha (PGC-1α). As a therapeutic approach, reinstating this mitochondrial biogenesis could enhance mitochondrial function and curb the progression of COPD [43].

### 3.5. Mitochondrial Quality Control Mechanisms

Mitochondrial quality control mechanisms, including mitophagy—the selective destruction of mitochondria by autophagy—and the mitochondrial unfolded protein response (UPRmt), are critical for ensuring mitochondrial integrity and preserving function [44]. In individuals with COPD, these quality control mechanisms face setbacks, leading to an accumulation of damaged mitochondria and subsequent mitochondrial dysfunction. Therefore, therapeutic strategies aiming at enhancing these quality control mechanisms could potentially preserve mitochondrial function, decelerating COPD progression [45].

Having a comprehensive understanding of the structural and functional changes the mitochondria undergo within the context of COPD is essential for identifying new potential therapeutic targets. This understanding can clear a path for the development of interventions focused on restoring mitochondrial health and stalling the progression of COPD.

## 4. Implications of Mitochondrial Dysfunction in COPD

Considered pivotal to the onset and growth of COPD, mitochondrial dysfunction’s influence ripples through multiple cellular and systemic avenues. This cornerstone role underscores the importance of a detailed comprehension of the havoc wrought by mitochondrial dysfunctions on COPD’s cellular machinery and systemic reactions, which could lead to potential therapeutic breakthroughs [46,47].

### 4.1. Cellular Signaling Pathways

At the cellular level, mitochondria command a central role by acting as signaling hubs associated with various critical processes like apoptosis, inflammation, and oxidative stress [48]. The emission of cellular distress signals from ailing mitochondria, including pro-apoptotic factors and reactive oxygen species (ROS), instigates apoptotic signaling cascades. This process accelerates tissue damage specific to COPD. Additionally, aberrations in mitochondrial task fulfillment disrupt intracellular signaling networks, inciting abnormal immune responses and fostering chronic inflammation emblematic of COPD [49].

### 4.2. Oxidative Stress and Inflammation

Acknowledged as the progenitors of heightened oxidative stress and inflammation seen in COPD, dysfunctional mitochondria indirectly instigate various disease aspects [46]. Inherent to ailing mitochondria is an unbalanced overproduction of ROS, creating an oxidative onslaught on various cellular entities. This domino effect feeds into causing inflammation by lighting up pro-inflammatory signaling routes, and mobilizing immune cells towards the lungs. The inflammation then further vaunts mitochondrial distress, setting off a cycle of disease potentiation.

### 4.3. Cellular Senescence and Tissue Remodeling

Mitochondrial dysfunction acts as a catalyst encouraging cellular senescence and tissue remodeling—potent COPD characteristics [50]. Excessively accumulated damaged mitochondria steer the cell towards senescence, ultimately stifling growth and impeding tissue repair mechanisms. Senescent cells, armed with pro-inflammatory cytokines and matrix metalloproteinases, contribute to COPD-induced tissue destruction and airway remodeling. Consequently, senescence hastened by mitochondrial malfunction may likely worsen lung function decline, increase COPD exacerbation susceptibility, and heighten overlying disease severity.

### 4.4. Metabolic Reprogramming and Muscle Dysfunction

Mitochondrial dysfunction swings the pendulum towards metabolic rearrangements, resulting in muscle dysfunction central to COPD [33]. Unhealthy mitochondria curb oxidative phosphorylation efficiency, thus causing a deficit in ATP supply and negatively impacting skeletal muscle strength and patient exercise tolerance. The metabolic reprogramming characterized by a glycolytic tilt and an overdependence on anaerobic metabolism may compound muscle wasting and functional impairments in COPD [51]. This implies that interventions aiming to rejuvenate mitochondrial health could aid in preserving muscle mass, enhancing exercise capacity, and improving overall patient wellness.

### 4.5. Exacerbation Susceptibility and Disease Progression

Evidence substantiates an intrinsic connection between mitochondrial dysfunction and an increased risk of exacerbations, including accelerated disease progression in COPD [52,53]. Dysfunctional mitochondria compromise the body’s cellular fortitude against external stressors, thereby heightening tissue damage during COPD exacerbations. Apart from exacerbation susceptibility, mitochondrial dysfunction also correlates with a rapid deterioration in lung function and accelerated COPD progression. Consequently, therapeutic blueprints looking to reinstate mitochondrial function might be crucial for minimizing the risks of COPD exacerbation and slowing down disease progression.

### 4.6. Genetic Predisposition, Environmental Influences, and Comorbidities

COPD development and progression are influenced by a combination of genetic predispositions, environmental exposures, and comorbid conditions. Genetic factors such as variations in genes encoding for proteins involved in antioxidant defenses, inflammatory responses, and tissue repair mechanisms can predispose individuals to COPD. Environmental factors, including long-term exposure to tobacco smoke, air pollution, and occupational hazards, significantly contribute to the disease’s complexity. Additionally, comorbidities like cardiovascular disease, diabetes, and metabolic syndrome often coexist with COPD, complicating its management and exacerbating disease outcomes [54,55]. Understanding the interplay between these factors and mitochondrial dysfunction is essential for developing comprehensive therapeutic strategies.

Understanding the comprehensive implications of mitochondrial dysfunction is vital to the landscape of COPD, serving as the building blocks for potential treatment breakthroughs. This comprehension could contribute to ushering in an era of effective interventions aiming to hinder disease progression and optimize clinical outcomes—perhaps a fresh dawn of hope for reversible cellular damage marked in COPD patients.

## 5. Therapeutic Targeting of Mitochondria in COPD (Table 2)

Mounting evidence points toward mitochondrial dysfunction as a crucial component in the pathology of COPD, making it a potential bullseye for therapeutic strategies [56]. Current investigations delve into a variety of therapeutic approaches intending to rejuvenate mitochondrial function and stall disease progression. These strategic lines of defense mainly focus on portfolio aspects such as the abatement of oxidative stress, amplification of mitochondrial biogenesis, and fortification of mitochondrial quality control mechanisms.
ijms-25-07780-t002_Table 2Table 2Biomarkers of Mitochondrial Dysfunction in COPD.BiomarkerClinical ImplicationsReactive Oxygen Species (ROS)Indicator of oxidative stress and disease severityMitochondrial DNA (mtDNA) DamageImpairment of mitochondrial functionMitochondrial Respiratory Chain Enzyme ActivityBiomarker of mitochondrial dysfunction and disease progression indicatorMitochondrial Biogenesis and Dynamics MarkersInsight into mitochondrial health and disease severityCirculating Mitochondrial ComponentsNon-invasive biomarkers for assessing mitochondrial dysfunction and monitoring disease progression

### 5.1. Antioxidant Therapy

Non-specific antioxidants have conventionally been deployed as a primary measure to curtail COPD’s oxidative stress, acting as neutralizing agents against ROS [57]. These include thiol-group compounds like N-acetylcysteine (NAC) and carbocysteine alongside dietary antioxidants like vitamins C and E. Encouraging results have been observed in animal models, indicating the beneficial effects of antioxidant therapies [58,59]. Real-world evidence gathered from clinical trials involving COPD patients, however, remains inconclusive. Despite small-scale trials suggesting NAC as potentially capable of reducing exacerbation frequency [60], larger studies have not corroborated such findings [61]. Furthermore, hyper-dosing non-targeted antioxidants could paradoxically inflict harm due to potential interference with physiological processes. Ambiguity around dosage guidelines and a lack of specificity collectively restricts their usage in clinical practice.

### 5.2. Mitochondrial Biogenesis Inducers

Pioneering a new potential COPD therapeutic avenue are mitochondrial biogenesis inducers, especially PGC-1α activators [43]. The fundamental role of PGC-1α in supervising mitochondrial biogenesis and function is to amplify the potential efficacy of therapeutic intervention across this axis. Experimental results from in vivo models exhibiting improved mitochondrial function and curbed inflammation through the pharmacological activation of PGC-1α inspire optimism in this field [62]. Currently, large-scale clinical trials probing the effectiveness of PGC-1α activators are underway. While still in exploratory stages, these studies may well pave the way for innovative, mitochondria-focused therapies.

### 5.3. Mitochondrial Quality Control Enhancers

Investigative interests lean towards enhancing mitochondrial quality control mechanisms as a tangible COPD combat strategy [63]. This typically involves encouraging the process of mitophagy, specifically targeting and eliminating damaged mitochondria. This approach could assist in purging dysfunctional mitochondria while maintaining cellular homeostasis within COPD-afflicted environments. In addition, sparking the mitochondrial unfolded protein response (UPRmt) might promote mitochondrial proteostasis, enhancing the mitochondrial functional response amid cellular stress. Ongoing preclinical research on the efficacy of mitophagy inducers and UPRmt activators in COPD is quite promising. It shapes a thrilling landscape filled with future therapeutic potentials awaiting their due acknowledgment in the field of COPD treatment.

### 5.4. Role of Peptides of Mitochondrial Origin

Recent research has uncovered the potential role of peptides of mitochondrial origin, such as MOTS-c and Humanin, in the context of COPD [64]. These peptides are involved in regulating metabolic processes, reducing oxidative stress, and modulating inflammatory responses. MOTS-c, for instance, has been shown to improve insulin sensitivity and reduce inflammation, which could be beneficial in managing COPD-related metabolic disturbances and inflammation [65]. Humanin has demonstrated protective effects against oxidative stress-induced cell damage, which may help preserve mitochondrial function and reduce tissue damage in COPD. Further studies are needed to explore the therapeutic potential of these peptides in COPD management.

## 6. Therapeutic Implications and Future Directions

Recognizing the pivotal role of mitochondrial dysfunction in COPD paves the way for complementing current treatment strategies with novel, mitochondria-targeted therapeutic interventions. The objective of these innovative therapies is to alleviate disease symptoms and impede disease progression [57]. Mitochondria—the powerhouses of the cell—are central to cellular energy production, the regulation of oxidative stress, and numerous cellular signaling pathways. Thus, their potential as a therapeutic target in managing COPD cannot be overemphasized.

Advancements in biomedical research have led to the formulation of mitochondria-targeted antioxidants (MTAs), specifically designed to penetrate the lipid bilayer of the mitochondria. This offers a direct, targeted approach to neutralize reactive oxygen species (ROS), the harmful byproducts of cellular metabolism, directly at their source [66]. These MTAs have trumped conventional, non-targeted antioxidants in preclinical studies by reducing apoptosis—a form of programmed cell death—and protecting against mitochondrial DNA (mtDNA) damage. This has brought MTAs to the forefront as promising therapeutic agents for COPD. As it stands, two clinical trials are examining the impact of a specific MTA, MitoQ, on COPD patients [67,68]. One of the substantial challenges in these trials relates to the determination of the most effective drug dosage and the establishment of consistent treatment protocols. This underscores the importance of standardizing these elements for the effective use of MitoQ in clinical practice and future COPD management.

Another potential therapeutic intervention in COPD entails modulating the dynamics of mitochondria, using specific compounds such as Dynamin-related protein 1 (Drp1) inhibitors and PTEN-induced kinase 1 (PINK1) activators. By influencing fundamental processes like mitochondrial fusion and fission, these compounds have the potential to restore the interconnectivity of the mitochondrial network, optimize the production of cellular energy—bioenergetics—and boost the survival of cells in COPD-affected lung tissues [39,66].

Moreover, enhancing mitophagy, the process of selectively removing damaged mitochondria, is emerging as an intriguing strategy to maintain mitochondrial health and prevent the buildup of dysfunctional organelles in COPD [69]. Compounds that stimulate mitophagy, such as rapamycin and urolithin A, may help eradicate damaged mitochondria, thereby reducing oxidative stress and mitigating cellular dysfunction—critical components of COPD pathogenesis.

A glimmer of hope for the future lies in the identification of novel mitochondrial biomarkers that could serve as unique, diagnostic indicators of mitochondrial dysfunction in COPD patients. These biomarkers could prove useful in early disease detection, and in observing responses to treatment over time, all while offering predictions of disease outcomes [70]. This could pave the way for personalized medicine approaches tailored to the individual needs of COPD patients.

Lastly, the complex interplay between mitochondria and other cellular pathways implicated in COPD pathogenesis deserves exploration. This includes pathways such as inflammation, autophagy—the cell’s recycling process, and cellular senescence—aging at the cellular level. Dissecting the intricate dynamics between mitochondrial dysfunction and these pathways could shed light on new therapeutic targets. Furthermore, it could offer fresh perspectives for developing innovative treatment strategies aimed at improving COPD management and patient outcomes [71].

### 6.1. Reactive Oxygen Species (ROS) and Oxidative Stress Markers (Table 2)

Mitochondrial dysfunction in chronic obstructive pulmonary disease (COPD) is linked with elevated levels of reactive oxygen species (ROS) and oxidative stress markers such as malondialdehyde (MDA) and 8-hydroxy-2′-deoxyguanosine (8-OHdG) [72]. The increase in ROS production by impaired mitochondria leads to oxidative damage to essential cellular components, further exacerbating inflammation and tissue damage characteristic of COPD. By measuring ROS and oxidative stress markers, researchers and clinicians may gain valuable insights into the severity of mitochondrial dysfunction and its relationship with disease severity in COPD [73]. Therefore, these variables could play an instrumental role in assessing, diagnosing, and monitoring the disease.

### 6.2. Mitochondrial DNA (mtDNA) Damage and Mutations

Mitochondrial DNA (mtDNA) damage and mutations are key indicators of mitochondrial dysfunction, and their detection can serve as potential biomarkers in COPD diagnosis and prognosis. It has been observed that oxidative stress and inflammation in COPD can cause significant mtDNA damage, resulting in impaired mitochondrial function and contributing to COPD progression. The regular monitoring and quantification of mtDNA damage and mutations provide a unique glimpse into the extent of mitochondrial dysfunction and its impact on COPD pathogenesis [74,75]. This not only illuminates the molecular underpinnings of the disease but also enables the design of personalized therapeutic interventions based on the severity of mtDNA damage or mutations in individual patients.

### 6.3. Mitochondrial Respiratory Chain Enzyme Activity

The evaluation of mitochondrial respiratory chain enzyme activity, particularly within complexes I, III, and IV, could prove to be a useful biomarker for indicating mitochondrial dysfunction in COPD. Any inhibition or reduction in the enzyme activity of these complexes indicates an impairment in mitochondrial respiration, leading to consequent deficits in ATP production and an upsurge in ROS generation—both crucial elements in COPD development and progression. The measurement of mitochondrial enzyme activity can provide an innovative diagnostic tool, offering possibilities to identify individuals suffering from mitochondrial dysfunction and to predict disease progression in COPD [76,77]. With such measurements, healthcare professionals can map the trajectory of the disease and adjust their treatment strategies accordingly, potentially leading to more effective management of COPD and improved patient outcomes.

## 7. Conclusions

The comprehensive study of the cellular and molecular processes that influence mitochondria in the context of COPD highlights the critical role mitochondrial dysfunction plays both in disease onset and progression. Key scientific insights reveal that mitochondrial dysfunction plays a fundamental role in promoting oxidative stress, inflammation, cellular aging, and tissue remodeling in COPD [78]. Changes related to mitochondrial structure and functionality, including modifications in their morphology, behavior, and energy production processes, could act as potential aggravators, amplifying disease severity and susceptibility to exacerbations.

Focusing therapeutics on addressing mitochondrial dysfunction exhibits tremendous potential in COPD treatment. Intervention strategies aimed at limiting oxidative stress, promoting the formation of new mitochondria, and initiating mechanisms to promote better mitochondrial quality control present possible pathways to decelerate disease progression and improve clinical outcomes in COPD patients [79]. By uncovering ways to restore mitochondrial health, therapeutic avenues are opened which could potentially ease inflammation, safeguard lung function, and enhance exercise tolerance in individuals with COPD.

Looking ahead, research should endeavor to unpack the complex mechanisms underlying mitochondrial dysfunction in COPD. Investigative inquiries into the interplay between mitochondrial dysfunction and other disease processes, such as inflammation, cellular aging, and metabolic shifts, will be central to the creation of targeted therapies [75]. Additionally, the detection of new biomarkers indicative of mitochondrial dysfunction and understanding their clinical implications may enable early diagnosis, inform prognosis, and monitor disease management in patients with COPD [80].

Bridging the gap between promising early-stage research findings and application in a clinical setting will necessitate concerted efforts in translational research. This requires the collaborative engagement of basic scientists, clinical experts, and pharmaceutical developers to translate preclinical interventions into robust, clinically effective therapies for COPD [81]. Further advancements in tailored therapeutic platforms, such as targeted drug delivery systems and gene editing technologies, could provide a springboard for precision medicine approaches in COPD management [82].

In conclusion, the comprehensive study of cellular and molecular processes influencing mitochondria in COPD highlights the critical role mitochondrial dysfunction plays in disease onset and progression. Incorporating insights into genetic predispositions, environmental influences, and mitochondrial peptides like MOTS-c and Humanin enhances our understanding. Key scientific insights reveal that mitochondrial dysfunction promotes oxidative stress, inflammation, cellular aging, and tissue remodeling in COPD. Focusing therapeutics on mitochondrial health shows tremendous potential in COPD treatment. Future research should aim to unpack the complex mechanisms underlying mitochondrial dysfunction, explore novel biomarkers, and develop targeted therapies. Translating preclinical findings into clinical practice requires collaborative efforts, paving the way for precision medicine in COPD management. The therapeutic targeting of mitochondrial dysfunction bears great promise for slowing COPD progression and improving clinical outcomes.

## Data Availability

No new data were created in this study. Data sharing is not applicable to this article.

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
