# Peer review of "Cellular and Molecular Biology of Mitochondria in Chronic Obstructive Pulmonary Disease"

_ijms, 2024, doi:10.3390/ijms25147780_

Round 1

Reviewer 1 Report

Comments and Suggestions for Authors

This article provides a detailed review of the role of mitochondrial dysfunction in COPD, considering various aspects of this topic. Given ongoing research on COPD and its treatment, the topic is timely and relevant, achieved by referencing the latest research. The authors mainly focus on mitochondrial dysfunction as a central aspect of COPD without sufficiently discussing other relevant factors, such as genetic predisposition, environmental influences, and comorbidities that contribute to the complexity of the disease. Moreover, the role of peptides of mitochondrial origin (MOTS-c, Humanin, etc.) and their potential impact on the development or course of COPD were not considered.

Because the article mainly focuses on studies confirming the role of mitochondrial dysfunction in COPD, it needs studies with different perspectives. Additionally, Some points and concepts are repeated throughout the article, which could be streamlined to improve readability and conciseness. Overall, the article is well organized, and sections flow logically from one topic to another. The arguments are coherent and complement each other effectively. However, on the one hand, dividing the article into subsections allows for better reception, and on the other hand, it reveals a small amount of information contained in individual subsections. In this situation, it is recommended that the article be rearranged for better reception.

Comments on the Quality of English Language

Minor editing of English language required

Reviewer 2 Report

Comments and Suggestions for Authors

Comments to the Authors

This is a review of the manuscript “Cellular and Molecular Biology of Mitochondria in Chronic Obstructive Pulmonary Disease”, by Li et al.

Dear authors,

I have no comments, the work is interesting, has novelty, is well-written and contributes to the field. Congratulations. I propose acceptance

Abstract

Your abstract is quite informative, however, it is a bit too long… I would try to reduce slightly the dimension~

Introduction

No issues detected with the introduction. It is informative and adequate.

Chapter 6.2. – careful with the font
